# Impact of Epidemic-Affected Labor Shortage on Food Safety: A Chinese Scenario Analysis Using the CGE Model

**DOI:** 10.3390/foods10112679

**Published:** 2021-11-03

**Authors:** Li Liang, Keyu Qin, Sijian Jiang, Xiaoyu Wang, Yunting Shi

**Affiliations:** 1Key Laboratory of Land Surface Pattern and Simulation, Institute of Geographic Sciences and Natural Resources Research, Chinese Academy of Sciences, Beijing 100101, China; liangl.16s@igsnrr.ac.cn (L.L.); jiangsj.17b@igsnrr.ac.cn (S.J.); shiyt.21b@igsnrr.ac.cn (Y.S.); 2University of Chinese Academy of Sciences, Beijing 100149, China; 3Key Laboratory of Ecosystem Network Observation and Modeling, Institute of Geographic Sciences and Natural Resources Research, Chinese Academy of Sciences, Beijing 100101, China; 4College of Resource Environment and Tourism, Capital Normal University, Beijing 100048, China; wangxiaoyu_simlab@163.com; 5Key Laboratory of 3D Information Acquisition and Application of Ministry, Capital Normal University, Beijing 100048, China

**Keywords:** COVID-19, food security, agriculture policy, labor, CGE

## Abstract

Human food safety should be given priority during a major public health crisis. As the primary element of agricultural production, labor tends to suffer the most during a period of public health concern. Studying the impact of epidemic-affected labor shortages on agricultural production, trade, and prices has important implications for food security. This study used a calculable general equilibrium model to study the changes in agricultural production, trade, and prices under different labor damage scenarios. The results showed that agricultural production was less affected under a scenario where the epidemic was controlled locally. The output of agricultural products decreased by about 2.19%, and the prices of agricultural products increased slightly. However, the nationwide output of agricultural products decreased by only 0.1%, and the prices remained largely stable. In the case of the spread of the epidemic, the output of agricultural products in the epidemic area decreased by 2.11%, and the prices of certain agricultural products increased significantly. For example, the price of vegetables increased by 0.78%, the price of pork increased by about 0.7%, and those of agricultural products in other parts of the country also increased slightly. Compared with the national spread scenario, the local outbreak scenario had a smaller impact on Chinese food security, indicating Chinese effective policy against the epidemic. Although the impact of labor shortage under the influence of the epidemic on China was relatively limited, and considering its stable food security, we should pay attention to the increase in the process of agricultural products and changes in agricultural trade in the epidemic area. The residents in the epidemic areas could not effectively obtain nutritious food, which affected their health. Thus, the government should also completely mobilize agricultural resources to ensure the nutrition safety of residents during major public health incidents.

## 1. Introduction

COVID-19 broke out in early 2020 and rapidly spread worldwide. Due to the highly contagious nature of COVID-19, as of 13 July 2021, a total of 188,012,550 cases had been confirmed worldwide, with 4,054,980 deaths. The continuous spread of COVID-19 has made matters worse. According to the statistics from Johns Hopkins University (USA), more than 10,000 people in India, the United States, Russia, Indonesia, and the United Kingdom are still getting infected with the novel coronavirus every day, implying that the global epidemic has not been completely brought under control and is still spreading. Thus, the situation of the epidemic is still grave; this may be one of the most serious public health security incidents in human history. COVID-19 has exerted both short-term and long-term negative impacts worldwide at several levels, likely leading to the loss of social welfare and humanitarian well-being, and posing a huge challenge to the world’s Sustainable Development Goals [1,2,3].

Food security refers to ensuring that everyone at all times can buy and afford enough food necessary for survival and health. It has four main connotations: availability, accessibility, utilization, and stability [4]. The global food chain needs well-functioning markets and dynamic linkages to move food from production to consumption more safely and cheaply. This is doomed to the higher vulnerability of the food system when encountering a global crisis [5]. The coronavirus pandemic has severely challenged the global food system and food security [6,7]. Therefore, it is highly necessary to seriously consider the related issues of food security under the crisis. Before COVID-19 became a global pandemic, 135 million people in 55 countries and regions were facing severe food insecurity [8]. With the outbreak and spread of the epidemic, previous studies have estimated that the number of people facing severe malnutrition in the world will double, accompanied by a surge in extreme poverty [9,10]. COVID-19 affects both the food demand and supply sides, and poses a serious threat to global food security. On the consumer side, a COVID-19 outbreak could lead to higher food prices. After the outbreak, the panic in the society will often cause residents to hoard a large number of supplies. This phenomenon may lead to a shortage of supply in the market [11,12]. Simultaneously, the shortage of production factors and unsmooth freight caused by the epidemic will also increase the production and transportation costs of several food products, especially fresh and perishable food [13,14]. Together, these two factors will inevitably lead to an increase in food prices. In addition, the impact of the epidemic on the global economy is likely to lead to a reduction in household income, making it impossible for the already vulnerable families to afford adequate and nutritious food [15,16]. On the supply side, measures to limit the further spread of the virus could disrupt food supply chains. Due to travel restrictions and labor shortages, several agricultural departments cannot deliver products or obtain raw materials in time [17]. Certain agricultural products sectors, such as the dairy, poultry, and egg industries, were affected by the closure of processing facilities, leading to the dumping of most of their products [18,19]. Some countries have adopted measures to restrict the export of agricultural products to ensure food security, which has a major impact on global agricultural trade, and further increases the risks of food insecurity and factory closures [7,20,21].

Labor, as one of the three major production factors, has suffered a more obvious impact during the epidemic; several studies have shown that the epidemic has significantly shortened the working hours of workers in many countries around the world, and has been accompanied by a large amount of unemployment [22,23,24]. The shock to the labor market will inevitably affect the stability of the food system. At present, several studies have shown that the shortage of labor affected by the epidemic has had many adverse effects on food supply and production. For example, factory shutdowns led to the loss of work for workers engaged in food processing, which in turn affected the food supply to a certain extent [25]; blockades and restrictions on cross-border movement caused seasonal labor and shortages, which in turn affected agricultural production [26,27,28]; mental and physical health problems caused by the pandemic will also reduce the working hours of skilled workers, which will affect agricultural production efficiency [29]. In addition, it is worth noting that studies have shown that labor shortages have different effects on food security in different regions. In developed areas, there are more efficient food supply chains [19], a higher degree of automation, and a high-level medical system, so it can relatively quickly adapt to this impact and adjust in time, but the relatively backward areas have very limited production efficiency and infrastructure levels, so it tends to show higher vulnerability in the face of labor shortages [30]. At the same time, in developing countries, the young labor force is increasingly absorbed by the secondary and tertiary industries, and the rural agricultural personnel are aging, which makes agricultural laborers in these countries more vulnerable to the impact of pandemic [31]. China is a country with a large population, and the Chinese government has always attached great importance to food safety. According to the China Rural Statistical Yearbook, China’s rural population in 2019 was approximately 552 million, accounting for 39.4% of the total population, of which 194 million were employed in the rural primary industry, accounting for 58.5% of the total rural employed population [32]. In the event of major public health emergencies, it is of great significance to ensure the safe production and supply of food. This study on the extent to which the sudden shortage of agricultural labor force will affect the production, trade, and price of food in China is conducive to having an understanding of the impact of public health emergencies on Chinese food safety, as well as to the government’s relevant policies to ensure food safety for its citizens [33,34,35,36]. This study uses a general equilibrium (CGE) model to simulate the impact of varying degrees of labor shortage on food production, trade, and prices in China, with an aim to provide a reference for governments’ strategies to ensure food safety in the face of major public health incidents.

## 2. Data and Methodology

### 2.1. General Framework

First, we collected and collated statistical yearbook data and multi-regional input-output data to construct a database for The Enormous Regional Model (TERM), including data for the 31 provinces and municipalities of the mainland. Second, we examined the agricultural scenario of China in detail by splitting it into 11 sectors and aggregated certain industrial sectors into one sector. Third, we set up two scenarios using TERM to analyze the effects of changes in a labor shortage in some provinces on the production of planting and animal husbandry, as well as the amount of change in the net export volume from different provinces and municipalities by 2019. In this study, “export” and “import” refer to trade in the domestic market, namely, the trade among 31 provinces and municipalities in China (Taiwan are not analyzed due to data unavailability).

### 2.2. TERM Model

TERM is a multi-regional CGE model that can be applied to a single country. It is constructed using the “bottom-up” method. It treats each region as a separate economy [37]. The economic entities are linked through trade and the flow of production factors. In addition, it can simulate the incomplete factor flow between regions and industries. The TERM model has two major advantages [38,39]. First, it has the ability to rapidly solve highly detailed large-scale models, such as industrial sector classification and regional division. Second, the TERM model uses a unique regional model database construction method. It can rapidly construct an accurate and credible regional model database despite the limited availability of regional information. The TERM model was developed by the Center for Policy Studies (CoPS) of the University of Victoria, Australia. The theory of the TERM model is similar to that of the general CGE model, which is a simplified representation of the economic system. The model simulates the conditions under which all agents use labor, capital, and land to produce goods. These commodities are traded between different regional markets and ultimately consumed by consumers. The system in the model maintains a dynamic balance between supply and demand [40].

### 2.3. Generating the TERM Database

We constructed a database of the TERM model based on the multi-regional input-output tables of 31 provinces in China in 2012 and the statistical data of 31 provinces in 2019.

Building the database can be roughly divided into three stages (Figure 1). The first stage is to sort and check the initial input data, followed by the use of the double scaling method (RAS) to check the balance between input and output. The second and most important stage is building the bottom-up database. This stage consists of three steps. The first step is to split users by destination; at this point, we are able to deduce regional demand for and supply of each commodity, domestic or imported. Differences between the two are accommodated by inter-regional trade, considered in the next steps. In the second step, we constructed the 2019 trade matrix based on the multi-regional input-output table of 42 departments of 31 provinces in China in 2012 [41] and the statistical yearbook data of each province in 2019. The third stage involved checking and aggregating regions and sectors. In this stage, we split the agricultural sector into 11 sub-sectors, including soybean and corn, wheat, rice and millet, vegetables, fruits, other crops, pigs, sheep and goats, other livestock, forest and fishing, and other meats. In addition, we aggregated certain industries into one sector, for example, coal mining and concentration products, oil and gas extraction products, metal mining and concentration products, non-metal mining and other mining, and concentration products are incorporated into the mining sector. Therefore, the database contained 47 sectors covering 31 provinces and municipalities of mainland China.

### 2.4. Scenarios

In classical economics, labor and capital are the primary factors of production, particularly in agriculture production. Agricultural production is a labor-intensive sector, and the labor force is experiencing a restructuring in rural China. In China, the middle-aged and elderly labor force accounts for the absolute majority of the rural labor force; these individuals are susceptible to several infectious diseases. Therefore, this study uses the COVID-19 outbreak in China as the background and considers fictional scenarios where agricultural labor force is damaged to varying degrees during major public health incidents.

In the first scenario, we assumed a limited infection scenario (LIS), assuming that the outbreak spread in the Hubei province and the spread of infectious diseases in the Hubei province had hit the agricultural labor force in the province by 2%. At the same time, the spread of the virus was controlled in the Hubei province due to effective measures and did not spread to other provinces. In the second scenario, we assumed that the virus had spread to the whole country (spreading infection scenario (SIS)). Under this scenario, the virus was widespread in the Hubei province, with limited transmission to Zhejiang, Hunan, Henan, and Guangdong provinces and no material impact on other provinces. Hubei’s labor force suffered a 2% loss, whereas the other four provinces suffered a 0.5% loss.

In TERM, there is an important difference between the identified scenario(s) and the basic scenario, which reflects the difference in the value of shock variables. Examples of these variables include the technical change in the labor force, technical change in the capital, technical change in the composition of primary factors, the number of households, the quantity of land, and total household consumption. This difference between scenarios should allow the investigation of the impact of a single variable on the system under study. The calculation of the model follows the economic assumption of complete employment, which also implies that we cannot obtain the desired effect by directly impacting the agricultural labor force because a direct impact on the agricultural labor force means that the labor force is transferred to other sectors, such as labor from planting industry moved to furniture manufacturing. The reality is that these laborers have to stay at home, or have lost the ability to work and have not been transferred to other sectors. Therefore, we achieved the desired effect by impacting the technical efficiency of labor, i.e., compared with the baseline scenario, there is a shortage of agricultural labor in the hypothetical scenario.

## 3. Results

### 3.1. Changes in Agriculture Output

After the outbreak of the epidemic, the impact of the labor force on agricultural product output can be intuitively observed from the agricultural product output change data of various provinces: Hubei province’s agricultural product output has been impacted by the labor force compared with the basic increase in other provinces, and its output has dropped significantly. Among them, the output of agricultural products in the Hubei province reduced by an average of 2.19%, with an average increase of 0.08% in other provinces.

The analysis of changes in the agricultural product output in the Hubei province under the impact of labor force showed that the output of vegetables, rice, and pigs increased, and that of other agricultural products, such as other livestock, other agricultural products, fruits, soybeans, wheat, and sheep, decreased (Table 1). The change in the yield of vegetables was highly evident with a percentage change of 0.07%, followed by the yield of rice with a percentage change of 0.06%. The yield of other livestock changed the least, with a percentage change of −0.01%. The data on the increase and decrease in agricultural products revealed that the output of basic household foods, such as vegetables and rice would generally maintain a slow-growth state, and that of other complementary foods would decline to varying degrees due to the impact of labor. For example, the severe decline in agricultural products is in sheep. Only in wheat, the percentages of change were −0.04% and −0.02%, respectively. Under the scenario of Hubei province being impacted by the labor force, the animal husbandry industry was severely impacted.

Next, we analyzed the changes after the labor shock from different provinces (Table 2). First, Hubei province, which was hit the hardest by the labor force, had a percentage change of −2.19% in the agricultural product output. Other provinces with labor shocks, such as Hunan and Shaanxi, increased their overall output by 0.09%, followed by Heilongjiang, Qinghai, Inner Mongolia, and Tibet; their agricultural output increased by 0.08%. Provinces less affected by labor, such as Sichuan, Guangdong, Yunnan, Guizhou, Jiangsu, and Shandong, reported an increase of 0.7% in agricultural output. Based on the impact of the labor force in the Hubei province, the output of agricultural products in various provinces across the country was analyzed. The output of regions with relatively rich agricultural resources or relatively developed economies, such as Hunan, Shaanxi, and Sichuan, still maintained a slow upward trend, such as Hunan, which had superior water, soil, and sunshine conditions. Its vegetable output increased the most, with a growth percentage of 0.20%. For areas with relatively scarce agricultural resources or underdeveloped economies, such as Qinghai, inner Mongolia, Tibet, and Beijing, the impact of the labor force in the Hubei Province was evident.

Analysis of changes in the agricultural product output across the country under the impact of labor showed that the changes in the output of agricultural products nationwide after the impact of the labor force were largely the same as those in the Hubei province alone. However, differences existed in changing trends among different provinces. We analyzed the changes after the labor shock from different crop types: the output of basic staple foods, vegetables, fruits, and pigs increased, and that of other agricultural products, such as other livestock, other crops, soybeans, and sheep, decreased. The change in the yield of vegetables was the most evident, with a percentage change of 0.12%, followed by the yield of pigs, with a percentage change of 0.09%. The smallest change was observed in the yield of fruits, with a percentage change of 0.002%. The most serious decline in the output of agricultural products was in other animal husbandry practices, such as sheep, with a decrease of 0.05%. Judging from the changes in the output of agricultural products nationwide after the impact of the labor force, the increase in the output of agricultural products across the country was largely in the daily basic food of the national residents, whereas other non-basic foods were affected by the national labor force and decreased.

We analyzed the changes after the labor shock from different provinces. Hubei province, which was hit the hardest, had a percentage change of −2.11% in agricultural product output. In addition to the negative growth in agricultural product output in Hubei Province, Zhejiang, Guangdong, Henan, and Hunan reported the following percentage changes in the agricultural output: −0.38%, −0.39%, −0.39%, −0.41%; provinces that were more affected by the labor force, such as Hunan, had a decrease of 0.41%, followed by Henan and Guangdong, with reported a decrease of 0.39%. The province with the least labor shock was Sichuan, where the output of agricultural products increased by 0.14%, followed by Yunnan, Guizhou, and Jiangsu, where the output of agricultural products increased by 0.15%. Based on the analysis of the national labor shock, the output of agricultural products in several provinces, such as Sichuan, Yunnan, Guizhou, Jiangsu, and Chongqing, where the primary industry accounted for a relatively small proportion of developed economies, the output still maintained a slow upward trend, such as the average output of agricultural products in Heilongjiang. With an increase of 0.18%, Shaanxi’s vegetable production was the highest, with a percentage increase of 0.32%. Areas with relatively scarce agricultural resources or underdeveloped economies, such as Zhejiang, Guangdong, Henan, and Hunan, were significantly affected by the national labor force.

Using the labor shock model, the total output of agricultural products in the country under normal conditions, the total output of agricultural products under the impact of the labor force in Hubei province, and the total output of agricultural products under the impact of the national labor force were obtained.

From the total output of agricultural products under different scenarios, it was found that regardless of the impact of the labor force in Hubei province or the whole country, the total output of agricultural products declined to varying degrees (Figure 2). The output of agricultural products was affected by labor significantly higher across the country than in the Hubei province. Soybean production showed the highest decline in total output by 0.49%, and the total output of other cereals that were less affected by the labor force decreased by 0.19%. Again, soybean production showed the highest decline at 0.18%, and the production of fruits that were less impacted by the national labor force declined by 0.06%. Under the impact of labor, the total output of agricultural products in the country was lower than the benchmark value under normal conditions. Soybeans mature later than other crops, and therefore the output declined significantly under the impact of labor.

### 3.2. Prices and Consumption of Agricultural Products

Under the scenario of only a 2% reduction in the labor force in Hubei province, the changes in the prices of commodities in Hubei province were the most affected (Figure 3). The prices of soybeans and corn increased by 0.31%, and those of wheat increased by 0.20%. Prices of rice and millet also increased. The price of vegetables and grapes increased by 0.73% and 0.25%, respectively. The prices of other crops increased by 0.26%. The prices of pork increased by 0.63%, those of mutton and goats increased by 0.11%, and those of other livestock increased by 0.41%. Rice and millet had the largest increase and the highest impact, followed by vegetables. This may be related to the fact that the production of these crops requires a large labor force and are the main necessities of life. In this scenario, with the exception of Hubei, Beijing, Shanghai, Jiangxi, and Hunan, other provinces and cities had relatively significant price increases compared to other provinces. Based on the status quo, we attributed the increase in prices in Beijing and other municipalities to their developed economy. The majority of the supply of necessities depends on other provinces; therefore, the impact of the reduction in labor in other provinces was more evident. The reason why provinces, such as Jiangxi and Hunan, were more affected by the reduction in the labor force of Hubei province was that these were adjacent to Hubei province, and had a large trade volume with each other.

Under the above scenario, the consumption of soybeans and corn in the Hubei province decreased by 0.11%, the consumption of wheat decreased by 0.05%, the consumption of rice and millet decreased by 0.34%, the consumption of vegetables decreased by 0.34%, the consumption of grapes decreased by 0.08%, and that of other crops dropped by 0.08%. The consumption of pork decreased by 0.29%, the consumption of mutton and goats decreased by 0.002%, and the consumption of other livestock decreased by 0.16%. The highest decline was recorded in the production of rice and millet, followed by vegetables. These data showed that the decline in consumption was largely proportional to the increase in prices.

In the second scenario, in addition to a 2% reduction in the labor force in Hubei, a 0.5% reduction in the labor force in Henan, Zhejiang, Guangdong, and Hunan was observed. Under this scenario (Figure 4), the prices of soybeans and corn in the Hubei province increased by 0.36%, wheat prices increased by 0.31%, rice and millet prices increased by 0.78%, vegetable prices increased by 0.78%, fruit prices increased by 0.31%, and prices of other crops increased by 0.30%. The prices of pork increased by 0.69%, prices of mutton and goat increased by 0.18%, and those of other livestock increased by 0.46%. Under this scenario, the price increase in these commodities in the Hubei province was higher than that in the first scenario. This may be because Hubei, Henan, Zhejiang, and other provinces usually trade in these commodities. The proportion of Hubei province’s vegetable trade as compared with other provinces was higher than that of rice and millet, which resulted in a higher increase in the prices of vegetables than those of rice and millet. Similarly, the commodity prices in the four provinces of Henan, Zhejiang, Guangdong, and Hunan increased significantly compared with those in the first scenario. In addition, the prices in other provinces increased by varying degrees.

In the second scenario, the reduction in the consumption of all commodities in the Hubei province increased compared with that in the first scenario. The largest decrease in consumption was also observed in vegetables, followed by rice and millet. The trend of consumption in the remaining provinces was similar to that of labor reduction in this scenario.

### 3.3. Changes in Regional Trade of Agricultural Products

Figure 5 shows the export of products from Hubei province to other provinces. Seven provinces had a trading volume of more than 200; among these, Guangdong’s trade volume was the highest. The trade volume of Hainan was the lowest. Among the nine products, vegetables had the highest trade volume, followed by pigs. The lowest trade was reported for wheat. In six crops, Henan province had the largest trade volume of rice, Shaanxi province had the largest trade volume of vegetables, Jiangsu province had the largest trade volume of fruits, and Guangdong province had the largest trade volume of soy, wheat, and other crops. In livestock, Jiangsu had the largest trade volume of pigs and other livestock, and Guangdong had the largest trade volume of sheep.

The decrease in the labor force in the Hubei province decreased the production of various products to different degrees. The export of vegetables from Hubei to other provinces was affected the most, with an average decline of 3.77%. The smallest change was observed in sheep, with an average reduction of 2.16%. The remaining products showed the following decrease in the order of large to small: pigs, soybean, fruits, wheat, other crops, other livestock, and rice. The average loss of crops was 3.1%, higher than the average reduction of 2.9% seen in livestock. In terms of trade from other provinces to Hubei, the trade of vegetables, fruits, and other livestock declined, whereas that of the other six increased. The highest decline was observed in vegetables (2.98%), and the highest increase was observed in pigs (0.5%).

Among the thirty provinces that traded with Hubei province for various commodities, nine commodities were affected to varying degrees. The export volume of wheat in Hunan and Jiangxi was affected the most, with an average decrease of 0.16%. The import volume of wheat in Sichuan was affected the most, with an average decrease of 0.24%. Similarly, the export and import volumes of rice in Shaanxi were the most affected, with an average decrease of 0.35% and 0.48%, respectively. The export volume of soybean in Hunan was impacted the most, with an average decline of 0.16%, whereas the import volume of soybean in Anhui was impacted the most, with a decline of 7.5%. The export of vegetables was the worst affected, with an average decrease of 0.27%. Vegetable import in Guangdong, Guangxi, Fujian, and Zhejiang was the worst affected, with an average decrease of 0.33%; however, the vegetable import increased in Hubei by 0.34%. The export volume of fruits in Shaanxi was the most affected with a decrease of 0.15%, whereas its import volume in Guangxi, Sichuan, Zhejiang, and Fujian was highly affected with a decrease of 0.24%. The export of other livestock from Shaanxi was greatly affected and decreased by an average of 0.22%, whereas its import from Sichuan, Guangxi, and Guizhou showed an average decrease of 0.28%. As for other crops, Shaanxi’s export volume was affected the most, with an average decrease of 0.18%, whereas Shandong’s import volume of other crops was affected the most, with an average decrease of 0.26%. With respect to sheep trade, Hunan and Shaanxi provinces were most affected by the outflow of trade volume, which decreased by 0.18%, whereas Anhui, Shandong, Yunnan, and Hainan provinces were the most affected by the outflow of trade volume, which decreased by 0.23%. The export of pigs reduced by 0.21% in Shaanxi, whereas its import in Sichuan, Yunnan, Guangxi, Guangdong, and Hainan reduced by 0.3%.

Product outflows were most affected in the Hubei province, where nine items decreased by an average of 3.2%, whereas vegetable outflow showed the highest decrease of 3.9% with the decline in the labor force. In terms of the inflow of goods, wheat inflow in 31 provinces decreased by 0.5% on average, among which Shandong province reported the highest decrease of 0.54%. The outflow of rice decreased by 0.67% on average in all provinces, with the largest decrease observed in Hubei (1.6%). The outflow of soybean decreased by 0.43% on average in all provinces, with Anhui reporting the highest decrease of 0.46%. The export of vegetables decreased by 0.54% on average in all provinces, led by Guangxi (0.62%). Fruit outflow decreased by 0.47% on average in all provinces and by 0.50% in Sichuan. An average reduction of 0.43% was reported for other crops, with Shandong reporting the maximum reduction of 0.47%. The outflow of pigs decreased by 0.53% on average; it decreased by 0.61% in Sichuan. Sheep outflow decreased by 0.47% on average and 0.49% in the Yunnan province. The outflow of other livestock dropped to 0.51% on average and was 0.55% in Sichuan.

## 4. Discussion

This study used the CGE model to simulate the impact of agricultural labor force damage on the output, prices, and trade of certain agricultural products in China during major public health events. Although these scenarios are hypothetical, and did not happen in reality, we still need to study the impact of possible scenarios, especially in China’s rural aging population, considering that the outbreak of extreme public health events may impact the world food security and social stability. We believe that this study can provide a more scientific understanding of the possible impact of public health emergencies on food safety, and help the government to formulate corresponding strategies to control the spread of the crisis. Robust results were obtained in both scenarios. When the epidemic was contained in the Hubei province, the decline in the output of agricultural products caused by the loss of labor was limited, and the national output decreased only by less than 0.2%. However, when the epidemic spread to several provinces and cities, the decline in the output caused by the loss of labor was highly significant, and the national output of certain agricultural products decreased by more than 0.5%. The decrease in the agricultural output was accompanied by an increase in the prices of agricultural products. The prices of agricultural products in the provinces affected by the epidemic increased significantly, among which Hubei province reported the highest increase, with the processing of certain products, such as pork, increasing by 0.8%. At the same time, the interregional trade of agricultural products was affected to varying degrees. The share of agricultural product flow in the Hubei province reduced significantly, and the agricultural product trade of Hunan, Jiangxi, and other provinces and cities was severely impacted.

For agriculture production, the two most important primary factors of production are labor and capital [42]. Compared to the high-level mechanized agricultural production in developed countries, agricultural production in China and other developing countries is still labor-intensive, and requires a large labor force to maintain agricultural production [43,44,45]. With the increasing urbanization, farmers have started switching to non-agricultural work to obtain higher labor remuneration. This phenomenon has reduced the rural labor force and increased the proportion of the elderly population [46,47,48]. It is simultaneously accompanied by relatively backward medical conditions in rural areas; these factors make the current agricultural labor force highly vulnerable to major public health incidents. The shortage of labor inevitably leads to the loss of yield; however, different kinds of agricultural products are affected differently. At present, the mechanization degree of major grain crops in China is relatively high, and the loss of labor force has a relatively small impact on the production of primary grain crops [36]. Unlike staple crops, the production of vegetables and fruits is majorly dependent on artificial cultivation, nursing, and picking, and the shortage of labor will greatly impact the production of these two kinds of agricultural products [14,49]. In addition, the production of meat will be hugely affected, especially the meat-processing industry, which not only needs more labor force but also requires one to work in a relatively closed environment, which is difficult to achieve under major public health events [50].

There has been research reported that the prices of agricultural products in China showed a slight upward trend during the epidemic [35]. This upward trend may be caused by several factors, such as the blocked supply chain, blocked international trade in agricultural products, and increasing international prices of agricultural products [51,52]. The results of our study show that labor shortage is also an important reason for the increase in agricultural product prices. With the decline in the output of agricultural products, their prices are bound to increase by varying degrees. Research shows that the price increase in different types of agricultural products is different. The price of staple foods had a small increase, while the price of vegetables, fruits, and meat increased more. This is the same as the results of previous studies [53]. Our results show that the price increase in agricultural products is relatively high (about 0.8%), which may not affect the food security of the region. However, we still need to consider that rising prices of vegetables, fruits, and meat will increase people’s food burden, especially among the poor, whose access to adequate and nutritious food will decrease [54]. This may change the dietary structure of residents to a certain extent, such as increasing the intake of staple foods and reducing the intake of meat and vegetables, which may lead to some social health problems, such as stunted youth development [55].

Considering the agricultural trade, provinces greatly affected by the epidemic showed a decrease in the outflow and an increase in the inflow. This was to be expected because the decline in agricultural production in certain provinces reduced the exports and imports from other provinces, thus, ensuring the province’s food supply. According to the proportion of grain production and consumption, provinces in China are divided into three types, namely, major grain-producing area, major grain-selling area, and major grain-balancing area. Ensuring the stable output of agricultural products in major grain-producing areas is of great significance to guarantee Chinese food security. In the event of a major public health event, if the agricultural labor force in major grain-producing areas is threatened, the grain supply in certain parts of China may become difficult to meet. From this point of view, it is essential to ensure a certain amount of grain reserves and design a reasonable import proportion for reducing the impact of emergencies.

In general, this study gives rise to the following policy implications. First of all, it is necessary to take strict measures to prevent the spread of the epidemic, both in the overall interest and in the local interest. The impact of the restricted epidemic on the entire country is limited, and regions not affected by the epidemic can help the affected areas through trade and other means to alleviate the degree of food and nutritional insecurity in the affected areas. Obviously, China’s successful epidemic prevention and control experience has shown us the benefits of preventing the spread of the epidemic. Second, providing food aid to the outbreak area is very necessary, our results showed that the outbreak area had improved agricultural production prices, and strict preventive measures against epidemics may make residents cannot easily obtain nutritious food. At this time, it is very important for the government to ensure the residents’ food and nutrition security by providing food reserves to the residents. Finally, unimpeded inter-regional trade is very important to food security, and inter-regional allocation of food resources can effectively relieve the pressure of food demand in areas severely affected by the epidemic. In the early stage of the epidemic, individual countries’ restrictions on agricultural exports cannot protect food security, but made the situation worse.

## 5. Conclusions

Based on the TERM model, this study designed two different infectious disease spread scenarios to simulate the impact of different degrees of labor impairment on agricultural production and food security. The results show that the degree and scope of the impact under different scenarios show obvious differences. Under the local diffusion scenario, the fluctuations in the output and price of agricultural products were relatively small, and almost only affected the provinces with close trade links. However, under the national spread scenario, the production and prices of agricultural products across the country changed significantly, and the inter-provincial trade flows have also suffered more, especially on labor-intensive and perishable foods. It can be seen that limiting the epidemic to local areas is of great significance to the food safety of both the whole country and the affected areas. Under the LIS scenario, the total agricultural output of Hubei province decreased by about 2.19% due to the loss of labor force, among which wheat, fruit, soybean, and corn harvest were most affected by a decrease of 2.76%, 2.73%, and 2.64%, respectively. The local labor loss had little impact on the national agricultural production, only resulting in a decline of about 0.2% of the national agricultural production. Agricultural production prices were also affected, with vegetable and pork prices rising the most, up 0.78% and 0.7%, respectively. In terms of trade flow, the local epidemic has led to an increase in the volume of trade in neighboring provinces, especially Jiangxi, Henan, and Shandong. Under the SIS scenario, the agricultural labor force in many provinces was damaged, which directly led to the decrease in agricultural production in many provinces. The total national agricultural production decreased by about 0.5%, among which the national soybean and corn production decreased by 0.49%, rice production by 0.39%, and vegetable production by 0.38%. Agricultural prices rose in many places, with Hubei province, the worst hit province, seeing the biggest rise. The SIS scenario is more serious than the LIS scenario, both in terms of the output and price of agricultural products, which highlights the necessity of epidemic prevention and control, and also proves that the Chinese government’s strong measures not only prevented the further spread of the epidemic, but also indirectly guaranteed the food and impact safety of the Chinese people.

## Figures and Tables

**Figure 1 foods-10-02679-f001:**
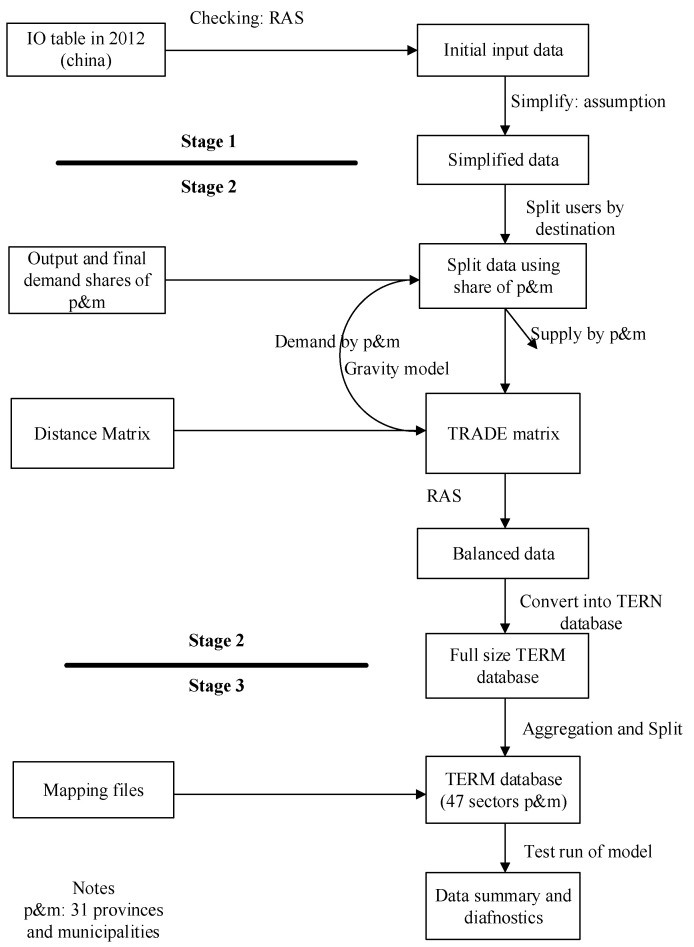
The three stages of constructing the TERM database used in the study.

**Figure 2 foods-10-02679-f002:**
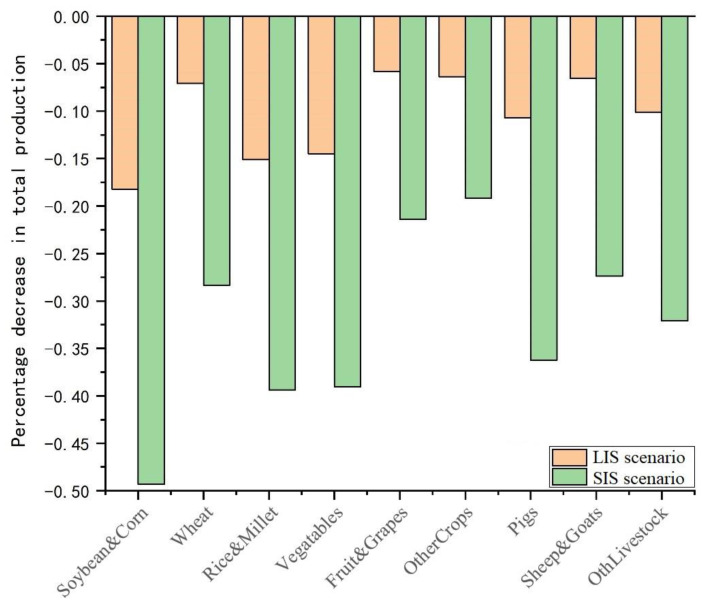
Change in national agricultural production output under different scenarios.

**Figure 3 foods-10-02679-f003:**
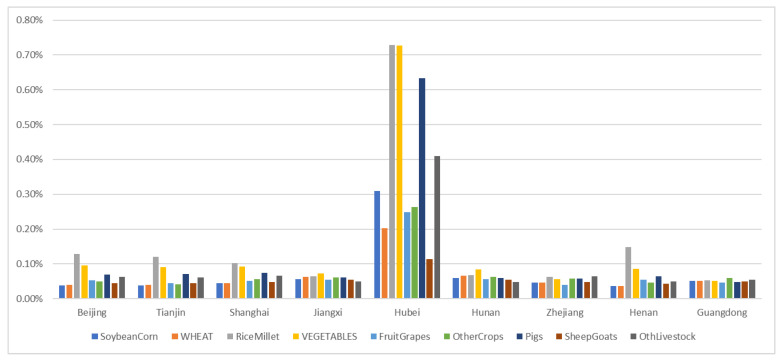
Under LIS scenario, price changes in Beijing, Tianjin, Shanghai, Jiangxi, Hubei, Hunan, Zhejiang, Henan, and Guangdong.

**Figure 4 foods-10-02679-f004:**
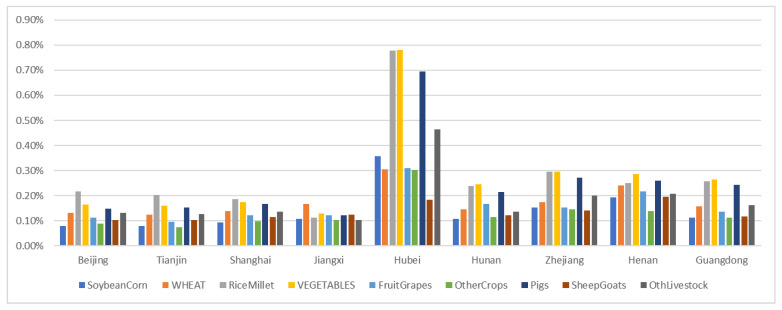
Changes in prices of various commodities in Beijing, Tianjin, Shanghai, Jiangxi, Hubei, Hunan, Zhejiang, Henan, and Guangdong under SIS scenario.

**Figure 5 foods-10-02679-f005:**
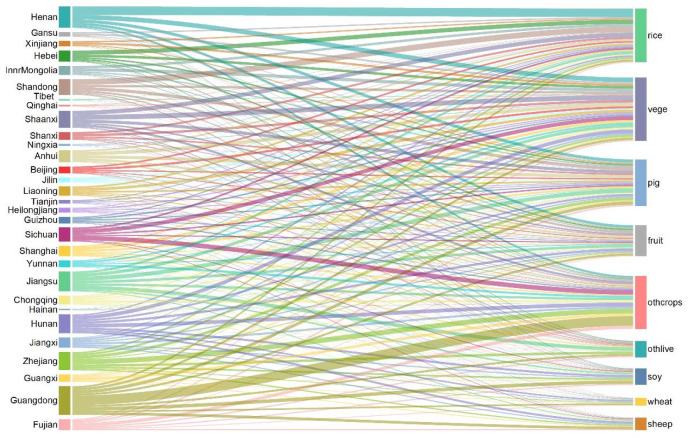
Changes in the amount of agriculture production exported from other provinces to Hubei provinces under SIS scenario.

**Table 1 foods-10-02679-t001:** Changes in agricultural production output in 31 provinces under LIS scenario (%).

	Vegetables	Rice & Millet	Pigs	Other-Livestock	Other-Crops	Fruit & Grapes	Soybean & Corn	Wheat	Sheep & Goats
Beijing	0.161	0.122	0.092	0.049	0.076	0.08	0.069	0.07	0.02
Tianjin	0.156	0.129	0.094	0.051	0.068	0.073	0.068	0.072	0.022
Hebei	0.154	0.127	0.094	0.041	0.072	0.082	0.066	0.064	0.014
Shanxi	0.176	0.126	0.095	0.043	0.085	0.077	0.055	0.061	0.005
InnerMongolia	0.179	0.122	0.094	0.039	0.09	0.082	0.061	0.066	0.015
Liaoning	0.16	0.12	0.091	0.045	0.093	0.077	0.059	0.075	0.016
Jilin	0.158	0.117	0.087	0.035	0.094	0.079	0.072	0.073	0.016
Heilongjiang	0.165	0.107	0.095	0.045	0.098	0.084	0.074	0.074	0.017
Shanghai	0.14	0.113	0.089	0.057	0.058	0.079	0.074	0.083	0.022
Jiangsu	0.114	0.091	0.085	0.067	0.064	0.075	0.062	0.055	0.027
Zhejiang	0.105	0.084	0.088	0.069	0.07	0.066	0.074	0.084	0.035
Anhui	0.14	0.108	0.105	0.04	0.074	0.081	0.059	0.062	0.015
Fujian	0.124	0.083	0.085	0.061	0.078	0.073	0.071	0.083	0.034
Jiangxi	0.143	0.11	0.104	0.043	0.079	0.072	0.078	0.075	0.026
Shandong	0.125	0.121	0.089	0.048	0.07	0.078	0.059	0.062	0.019
Henan	0.154	0.143	0.096	0.037	0.07	0.078	0.061	0.065	0.014
Hubei	−2.14	−1.59	−1.93	−1.55	−2.49	−2.73	−2.64	−2.76	−1.92
Hunan	0.156	0.111	0.116	0.033	0.084	0.076	0.086	0.079	0.026
Guangdong	0.104	0.073	0.062	0.05	0.061	0.073	0.075	0.09	0.038
Guangxi	0.136	0.101	0.083	0.048	0.069	0.068	0.066	0.078	0.03
Hainan	0.15	0.108	0.093	0.057	0.065	0.076	0.071	0.078	0.026
Chongqing	0.129	0.1	0.094	0.053	0.091	0.07	0.066	0.07	0.023
Sichuan	0.126	0.095	0.091	0.042	0.068	0.066	0.06	0.049	0.019
Guizhou	0.124	0.101	0.07	0.042	0.084	0.068	0.057	0.067	0.022
Yunnan	0.123	0.085	0.082	0.051	0.077	0.071	0.063	0.065	0.018
Tibet	0.16	0.13	0.088	0.047	0.082	0.073	0.074	0.072	0.017
Shaanxi	0.199	0.145	0.111	0.045	0.068	0.078	0.064	0.052	0.003
Gansu	0.173	0.123	0.086	0.04	0.079	0.077	0.061	0.061	0.015
Qinghai	0.171	0.118	0.092	0.043	0.085	0.076	0.084	0.072	0.016
Ningxia	0.174	0.138	0.095	0.045	0.055	0.072	0.065	0.059	0.017
Xinjiang	0.167	0.128	0.096	0.029	0.085	0.076	0.049	0.051	0.021

**Table 2 foods-10-02679-t002:** Changes in agricultural production output in 31 provinces under SIS scenario (%).

	Vegetables	Rice & Millet	Pigs	Other Livestock	Other Crops	Fruit & Grapes	Soybean &Corn	Wheat	Sheep & Goats
Beijing	0.277	0.225	0.212	0.234	0.182	0.108	0.129	0.136	0.07
Tianjin	0.271	0.233	0.216	0.238	0.171	0.112	0.113	0.132	0.078
Hebei	0.264	0.208	0.218	0.222	0.192	0.096	0.124	0.135	0.066
Shanxi	0.298	0.24	0.215	0.223	0.182	0.1	0.142	0.116	0.049
InnerMongolia	0.307	0.23	0.213	0.236	0.19	0.089	0.148	0.124	0.067
Liaoning	0.278	0.227	0.206	0.246	0.18	0.102	0.153	0.118	0.066
Jilin	0.277	0.215	0.198	0.244	0.185	0.085	0.155	0.141	0.064
Heilongjiang	0.287	0.235	0.183	0.251	0.199	0.098	0.161	0.147	0.064
Shanghai	0.264	0.244	0.189	0.249	0.194	0.115	0.093	0.144	0.074
Jiangsu	0.2	0.208	0.149	0.188	0.181	0.138	0.107	0.127	0.084
Zhejiang	−0.318	−0.322	−0.234	−0.442	−0.469	−0.327	−0.341	−0.557	−0.368
Anhui	0.24	0.209	0.17	0.204	0.189	0.097	0.123	0.12	0.064
Fujian	0.24	0.203	0.148	0.244	0.182	0.129	0.125	0.142	0.098
Jiangxi	0.255	0.228	0.183	0.228	0.175	0.099	0.125	0.148	0.083
Shandong	0.222	0.22	0.212	0.209	0.187	0.108	0.119	0.122	0.068
Henan	−0.306	−0.294	−0.269	−0.469	−0.479	−0.32	−0.523	−0.516	−0.373
Hubei	−2.03	−1.81	−1.53	−2.6	−2.62	−1.5	−2.45	−2.57	−1.86
Hunan	−0.295	−0.309	−0.25	−0.489	−0.537	−0.304	−0.513	−0.559	−0.401
Guangdong	−0.332	−0.284	−0.257	−0.451	−0.515	−0.323	−0.352	−0.575	−0.376
Guangxi	0.26	0.199	0.174	0.23	0.167	0.112	0.11	0.129	0.085
Hainan	0.292	0.236	0.192	0.232	0.188	0.134	0.111	0.142	0.075
Chongqing	0.214	0.189	0.161	0.217	0.168	0.104	0.141	0.123	0.075
Sichuan	0.216	0.195	0.156	0.18	0.163	0.094	0.109	0.118	0.061
Guizhou	0.222	0.194	0.176	0.215	0.163	0.095	0.132	0.109	0.067
Yunnan	0.22	0.184	0.147	0.211	0.171	0.113	0.124	0.122	0.059
Tibet	0.279	0.242	0.212	0.234	0.177	0.108	0.134	0.14	0.067
Shaanxi	0.317	0.254	0.224	0.192	0.182	0.098	0.112	0.125	0.048
Gansu	0.292	0.214	0.215	0.206	0.179	0.097	0.132	0.121	0.067
Qinghai	0.295	0.24	0.211	0.235	0.183	0.101	0.139	0.153	0.068
Ningxia	0.293	0.236	0.221	0.204	0.168	0.097	0.094	0.128	0.069
Xinjiang	0.291	0.245	0.216	0.177	0.179	0.072	0.14	0.101	0.076

## Data Availability

Not applicable.

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
