# Peer review of "Impact of Epidemic-Affected Labor Shortage on Food Safety: A Chinese Scenario Analysis Using the CGE Model"

_foods, 2021, doi:10.3390/foods10112679_

Round 1

Reviewer 1 Report

The literature review is incomplete. Please refer to more recent scientific publications which are related by topic. 

Please consider whether it is necessary to refer to the same source more than once in the same paragraph: "Consequently, people’s food burden will increase, especially in the poorer groups, and their access to food will decrease [27]. Our results showed that the extent of the increase in the prices of agricultural products is high (about 0.8%), which may not affect the region’s food security. However, we still need to consider that this price may change the dietary structure of residents, such as more intake of staple crops and reducing on meat and vegetable intake. This dietary change may lead to a series of social health concerns, such as teenager dysplasia [27]."

Please pay attention to the correctness of some entries, e.g. "Covid-19". Please also adapt the manuscript to the journal's requirements, e.g. References. 

Are some conclusions too obvious? Maybe it's worth trying to write them down differently? E.g.: "The results showed that the decrease in the agricultural labor force due to the epidemic reduced agricultural production, increased the prices of several commodities, and changed the agricultural trade flow."

What was the reason for the decline in total national agricultural production? Pandemic or weather conditions? Or maybe there are other reasons. The question is whether a decline at such a low level is of any importance and is it possible to identify its causes? "The total national agricultural production decreased by about 0.5%, among which the national soybean and corn production decreased by 0.49%, rice production by 0.39%, and vegetable production by 0.38%."

Author Response

Dear reviewer 1,

On behalf of my co-authors, we thank you very much for giving us an opportunity to revise our manuscript. We greatly appreciate your positive and constructive comments and suggestions on our manuscript.

We have studied your comments carefully and made revision in the places marked in red in the paper.

  1. We have made major revisions to the introduction that are close to a rewrite, adding many references related to the topic of the paper.
  2. We corrected the irregular quotation in the manuscript.
  3. We carefully checked and corrected the incorrect entries in the manuscript.
  4. We revised the obvious part of the description of the paper and instead described the research results and some possible policy implications around the purpose of the research.
  5. We have carefully considered your last comment, and added necessary notes to the manuscript. The loss of agricultural output and the rise of agricultural product prices mentioned in the article are caused by labor shortage and have nothing to do with other factors, which is determined by the setting of general equilibrium model. We only shocked labor factors without changing other factors.In order to avoid misunderstanding, we also added the description of the qualification conditions in the discussion, pointing out that the changes in output and price in the article are from the shortage of labor.

We have tried our best to revise our manuscript according to the comments. Attached please find the revised version, which we would like to submit for your kind consideration. We would like to express our gratitude again to you for the comments on our paper. I am looking forward to hearing from you.

Yours sincerely,

Liang Li

Reviewer 2 Report

The article deals with important issues related to food security. Changes in agricultural production, trade and prices in China were examined under different labor damage scenarios. However, the very concept of food security has not been defined.

The article repeats the entire paragraph twice:

Lines 84-87: „Several countries, including China, the USA, and Italy, had to ask citizens to stay at home to stop the spread of the virus and reduce the number of deaths. Although activity limitation has controlled the rate of virus transmission, it has incurred huge economic losses and unemployment; for example, the US unemployment rate for the second quarter of 2020 could exceed 30% [12]”.

And lines: 96-99

„Many countries like China, the USA, and Italy had to keep citizens stay at home to stop the spread of the virus and reduce the number of deaths. Although activity limitation has controlled the rate of virus transmission, it caused a lot of economic losses and unemployment; for example, the US unemployment rate for the second quarter of 2020 could exceed 30% [12]”.

Line 432: The reference to a bibliography is inappropriate.

The manuscript should be supplemented with figures on employment in the agri-food sector. Greater attention should be paid to the importance and characteristics of the labor factor.

In my opinion, figure 5 is hardly legible. Maybe it is better to make a compilation in a table.

The added value of the conducted research and further research directions should be indicated. There are also no recommendations regarding agricultural policy.

The reference to literature is rather short, you can enrich the bibliography overview, especially in the discussion section.

Author Response

Dear reviewer 2,

On behalf of my co-authors, we thank you very much for giving us an opportunity to revise our manuscript. We greatly appreciate your positive and constructive comments and suggestions on our manuscript.

We have studied your comments carefully and made revision in the places marked in red in the paper.

  1. We added the definition of food security in lines 49-51.
  2. We deleted the duplicated part in 96-99 and reorganized the narrative in that part.
  3. We have revised the format of the reference in line 432.
  4. We supplemented with figures on employment in the agri-food sector in line 217-221.
  5. We have added a paragraph to the discussion section to illustrate the policy implications of this research
  6. The introduction and discussion sections have been significantly revised and about 30 new references have been added
  7. We have seriously considered your opinion on Figure 5 and tried to express it in a table. However, due to the large amount of information involved in this chart, the form of the table is very ugly and difficult to edit, so we did not modify it.

We have tried our best to revise our manuscript according to the comments. Attached please find the revised version, which we would like to submit for your kind consideration. We would like to express our gratitude again to you for the comments on our paper. I am looking forward to hearing from you.

Yours sincerely,

Liang Li

Round 2

Reviewer 1 Report

The manuscript still requires minor corrections, e.g., the required elements are missing: 

  • Author Contributions
  • Institutional Review Board Statement
  • Informed Consent Statement
  • Data Availability Statement
  • Conflicts of Interest